# Photodynamic Therapy in the Treatment of Vulvar Lichen Sclerosus: A Systematic Review of the Literature

**DOI:** 10.3390/jcm10235491

**Published:** 2021-11-23

**Authors:** Agnieszka Gerkowicz, Paulina Szczepanik-Kułak, Dorota Krasowska

**Affiliations:** Department of Dermatology, Venerology and Paediatric Dermatology, Medical University of Lublin, 20-081 Lublin, Poland; agerkowicz@wp.pl (A.G.); dor.krasowska@gmail.com (D.K.)

**Keywords:** vulvar lichen sclerosus, VLS, photodynamic therapy, PDT

## Abstract

Vulvar lichen sclerosus (VLS) is a chronic inflammatory disease involving the genital skin and mucous membrane. Patients exhibit focal atrophy and destructive scarring, with an increased risk of malignant transformation. Due to objective symptoms as well as subjective complaints, patients with VLS experience emotional distress, lowered mood, and sexual dysfunction, which is reflected in impaired health-related quality of life. Thus, the necessity of implementing appropriate therapy at the earliest possible stage of the disease in order to avoid serious complications is highlighted. We presented the systematic review of available literature, performed with MEDLINE, Cinahl, Central, Scopus, and Web of Science databases. We identified a total of twenty relevant studies which indicate that photodynamic therapy (PDT) is a valuable therapeutic modality in the treatment of VLS.

## 1. Introduction

Vulvar lichen sclerosus (VLS) is a chronic inflammatory disease involving the genital skin and mucous membrane. The first description of lichen sclerosus lesions became known in 1887 [1]. Initially, no uniform terminology existed, with many synonyms being used. Nowadays, the term “vulvar lichen sclerosus” is widely applied [2]. VLS is a chronic disease with a pattern of recurrent lesions. Patients exhibit focal atrophy, destructive scarring, and associated functional impairment, with an increased risk of malignant transformation [3]. Worth mentioning, lichen sclerosus may also interest other body areas, and the genital area of the opposite sex [4].

The worldwide prevalence of lichen sclerosus ranges from 0.1% to 1.67% [5]. Of note, the exact prevalence of VLS has not been established and is probably underestimated, particularly in young women [6]. Several papers report a bimodal onset of disease, that is, in prepubertal children and postmenopausal women [2,7]. However, recent studies suggest that VLS may also affect some other age groups [5]. In terms of incidence, elderly women predominate (3%), while lichen sclerosus affects about 0.07% of men [8,9]. The ratio of men to women varies between 1:3 and 1:10, and only rarely is an equal split observed [9]. Approximately 0.3% of children are estimated to be involved [10]. The etiology of VLS remains unknown, but several mechanisms have been investigated [11]. Studies suggest a multifactorial origin of the etiology, such as autoimmune mechanisms, genetic predisposition, association with viral diseases, trauma, chronic irritation, and endocrine disorders [3,12].

In the early stages of VLS, well-demarcated, thin, glistening, ivory-white areas are seen, usually located in the labia minora, vaginal introitus, and fork. Tenderness and fragility, characteristic features of VLS, lead to the development of erosions, fissures, purpura, and petechiae. Fissures are particularly common between the clitoris and urethra, in the interlabial sulci, and at the base of the posterior fourchette. In addition, these lesions often occur during sexual intercourse or aggressive physical examination. With time, late complications resulting from the development of atrophic changes and scarring may appear. Mucocutaneous lesions are accompanied by itching, which is particularly severe at night and therefore worsens the quality of sleep, as well as intense pain, dyspareunia, or even apareunia, and impeded urinary flow. Furthermore, when the perianal area is involved, which occurs in about one-third of women, problems with defecation may occur [2,8,11,13]. Of concern is the association between VLS and subsequent vulvar squamous cell carcinoma (SCC). In their systematic review, Spekreijse et al. reported that the absolute risk of developing SCC in women with VLS varied between 0.21% and 3.88%. Contributing factors to this risk included age, presence of vulvar intraepithelial neoplasia, long medical history of VLS, delayed diagnosis of VLS, and only partial compliance in terms of use of the topical treatment [14].

According to the guidelines published, the diagnosis of VLS is based on clinical and histopathologic features. In addition, it is necessary to exclude several other disorders [9,15]. Noteworthy is the possibility of employing dermoscopy as a non-invasive diagnostic method [16,17].

Due to objective symptoms as well as subjective complaints, patients with VLS experience emotional distress, lowered mood, and sexual dysfunction, which is reflected in impaired health-related quality of life [18,19]. Thus, the necessity of implementing appropriate therapy at the earliest possible stage of the disease in order to avoid serious complications is highlighted.

The first-line treatment and the standard of care for VLS is ultrapotent topical corticosteroids (TCSs). According to the British Association of Dermatologists (BAD) guidelines published in 2018, the recommended treatment for VLS is the use of clobetasol propionate ointment 0.05% in one fingertip unit once daily for a month, then in the second month every other day and in the third month twice a week, in combination with a soap substitute and barrier preparation [20]. Topical corticosteroids are also the mainstay of treatment for exacerbations in the course of the disease [20]. The topical calcineurin inhibitors (TCIs), tacrolimus and pimecrolimus, are second-line agents that are less effective than clobetasol propionate in the treatment of VLS [21]. Other treatments for VLS include cyclosporine, phototherapy (narrowband UVB, UVA1) and photochemotherapy (PUVA), oral retinoids, and methotrexate [22]. In addition, in patients with lesions refractory to TCS, encouraging results have been obtained after using adipose tissue-derived stem cells and/or platelet-rich plasma [23]. There are also reports of the use of three energy-based methods: photodynamic therapy (PDT), high-intensity focused ultrasound (HIFU), and fractional CO_2_ laser (FxCO_2_) [21]. Surgical procedures are often required for late complications of scarring and adhesions [21]. However, the current evidence for the efficacy of the mentioned therapies in the treatment of VLS is poor. Further research is needed to establish recommendations for these therapies.

The role of photodynamic therapy (PDT) in the treatment of VLS is of particular interest, given its high efficacy and good cosmetic outcome. According to evidence-based guidelines, it is a method with a clinical benefit, especially in terms of subjective complaints related to the presence of skin lesions. Moreover, PDT should be considered in cases refractory to standard treatment. However, the patient should be informed about the need to carry out several treatments, which contributes to a long treatment time, and about the discomfort that may be experienced during the treatments [9].

PDT is a treatment consisting in the use of a photosensitizing chemical substance to cause phototoxicity [24]. This method is based on the interaction of three non-toxic agents: the light of appropriate wavelength, photosensitizer, and oxygen, which collectively lead to selective photooxidation of lesional tissues without damage to the surrounding healthy skin [25]. In this paper, we discussed the use of PDT in VLS by performing a systematic review of the available literature.

## 2. Materials and Methods

The systematic review of the literature was performed with MEDLINE, Cinahl, Central, Scopus, and Web of Science databases complementary to PRISMA (Preferred Reporting Items of Systematic Reviews and Meta-Analyses) protocol. The inclusion criteria for considering studies for the review based on PICOS structure comprised the population of patients diagnosed with VLS undergoing PDT, English language publications, clinical trials, and publications from 2000–2021. The criteria of exclusion included review articles, animal studies, as well as comparative, immunological, or histopathological studies. The databases were searched using the relevant MeSH terms: “vulvar lichen sclerosus” and “photodynamic therapy”. The search was performed in August 2021, with the last day to access the databases of 13 August 2021.

## 3. Results

The initial search revealed 182 results. After applying criteria of exclusion and inclusion, the database search revealed 20 records (MEDLINE *n* = 20, Cinahl *n* = 0, Central *n* = 0, Scopus *n* = 0, Web of Science *n* = 0) published in 2005–2021 submitted for the further analysis (Figure 1).

The studies included in the systematic review based on their publication date are presented in Table 1.

### 3.1. Characteristics of Included Studies

The majority (*n* = 11) of the included studies were prospective, with the remainder involving case series (*n* = 5) and case reports (*n* = 4). Only one study comprised a controlled cohort [37]. The number of reported patients ranged from 1 up to 102.

The study group consisted of women between the age range of 9 and 85 years (52 on average) with a clinically and histopathologically confirmed diagnosis of VLS. Notably, Cao et al. described the treatment of the hyperkeratotic form of VLS [42]. The duration of the disease ranged from 3 months to 29 years. Patients had a history of receiving local, systemic, or surgical treatment that did not lead to significant clinical improvement. The follow-up time for patients was under 12 months in the vast majority of studies.

### 3.2. Photodynamic Therapy Parameters of Included Studies

For the majority of studies (*n* = 17), 5-aminolevulinate (5-ALA) was used as the photosensitizer. The concentration range of 5-ALA varied, being 20% (*n* = 10), 10% (*n* = 3), and 5% (*n* = 4). Of particular interest, Zawiślak et al. described the use of a bioadhesive patch system in which the estimated dose of 5-ALA was 38 mg/cm^2^. In addition, some investigators used 5-ALA with the addition of dimethyl sulfoxide (DMSO) at a concentration of 2% or 20% (*n* = 4), mainly due to the increased efficacy of the treatments resulting from the higher penetration of 5-ALA into deeper skin layers [36,37,40,45]. Methyl aminolevulinate (MAL) was employed in two studies [30,34]. The duration of the maintenance of the photosensitizer on the skin ranged from 115 min to 6 h, with a clear dominance of the 3-h period (*n* = 11).

Regarding the light source, red light was used in 19 studies, and one study described the use of green light. For most studies (*n* = 18), the range of wavelengths used coincided with the preferred therapeutic window of 600–800 nm for PDT. In 14 papers, the wavelength oscillated between 630–635 nm. Notably, Olejek et al., in a study on a large group of patients (*n* = 100), applied two light sources [37]. Group I received treatment with red light (DIOMED 630 nm) and group II with a combination of visible light and water-filtered infrared A light (PhotoDyn^®^ 750 Heine.Med GmbH & Co. KG Müllheim, Germany, 580–1400 nm). Both groups experienced a significant reduction in the intensiveness of subjective complaints, with no significant difference between them [37].

There was quite a wide variance in the treatment parameters, that is, dose and intensity. The doses fluctuated between 9 and 180 J/cm^2^, with a range of 100–150 J/cm^2^ in eight studies. For light intensity, a minimum of 40 and a maximum of 700 mW/cm^2^ were employed. In eight studies, the light intensity used covered 80 mW/cm^2^. The number of treatments performed varied between 1 and 10, with the highest number of three procedures (*n* = 5). Moreover, PDT treatments were performed at different intervals. The majority were 2 weeks (*n* = 9), and less frequently 1 week and 4 weeks (*n* = 4, *n* =3, respectively).

### 3.3. Main Outcomes

Special concern was given to subjective symptoms, especially pruritus, which is considered as a major discomfort for patients with VLS. Sixteen studies have described the efficacy of PDT in the resolution of pruritus. The considerable reduction of itching as measured by VAS or VRS scales was reported in eight publications [28,29,31,34,35,38,40,41]. Other studies indicated high therapeutic efficacy of PDT in relieving pruritus, in some cases of a gradual nature with subsequent treatments [27,30,32,33,37,42,43].

Because of the location of the lesions, VLS favors the development of sexual dys-function. Seven studies have addressed this issue [27,32,33,38,40,41,45]. Only one study assessed patient satisfaction with treatment [38]. Lan et al. noted that 9/10 patients were highly satisfied, and 1/10 patients was satisfied [38].

In addition to the effect on subjective symptoms, there have been reports of relief of objective features after PDT. Romero et al. and Osiecka et al. described healing of erosions [27,33,35]. However, in a study by Romero et al. only the external erosions were cleared, whereas the deeper ones remained [27]. Other cases of clinical enhancement were due to a reduction in the size of VLS foci or their complete cessation [38,42,43,44]. Of note, Li et al. reported a significant reduction in the total objective score 6 months after PDT, which included symptoms such as leukoplakia, erythema, hyperkeratosis, purpuric lesions, and itching-related excoriations [41]. There are also reports of poor effects of PDT on the clinical features of VLS, despite satisfactory resolution of subjective complaints. Sotiriou et al. described a mediocre improvement in terms of clinical signs (hyperkeratosis, atrophy, induration, depigmentation) in 9/10 women receiving PDT and a complete lack of improvement in 1/10 patients [28]. In addition, Imbernón-Moya demonstrated that the evaluation of VLS lesions regarding clinical and morphological aspects was similar before and after PDT [34].

Histopathological examinations after PDT were performed in four publications [29,31,44,45]. The study by Sotiriou et al. emphasized the effect of PDT on subjective symptoms only, as no improvement in histopathological findings was found [29]. Zawiślak et al. also showed no histopathological improvement [31]. However, based on immunohistochemical evaluation, they found an increase in the apoptotic index, which allowed them to conclude that PDT has an effect by inducing apoptosis within the treated tissues [31]. Zhang et al. observed resolution of chronic inflammation and acellular collagen bundles and improvement in the epithelial vacuolar degeneration [44]. However, in the study by Zielińska et al., histological improvement was found in only 2/73 of the treated women [45]. Different diagnostic tools were used to objectively evaluate the effectiveness of treatment, including: dermoscopy, photodynamic diagnostics, or vulvoscopy [32,36,37,43].

### 3.4. Adverse Effects

The observed overall good tolerability of PDT treatment and short persistence of complications are worth mentioning. In none of the studies was the continuation of treatment abandoned due to complications. However, adverse effects have been reported in almost every publication, except for the study by Olejek et al. in which no complications were noted [37]. Only Vano-Galvan described the patient’s reluctance to restart treatment when symptoms worsened at 4 months after the end of treatment. The reason for this was the intensity of the pain suffered during PDT and poor tolerance of local anesthesia [30].

The predominant complaint mentioned in the studies (*n* = 12) was mild to intense pain, which sometimes exceeded the duration of the procedure but was short-lived. Notably, not all studies reported the use of anesthesia preoperatively or during the procedure. Vano-Galvan et al. used intralesional injections of mepivacaine 2% before each procedure, and Zawiślak provided oral co-codamol and topical lignocaine gel [30,31]. Zhang et al. administered oral analgesics and lignocaine in 2 of 28 and 14 of 30 patients, respectively [40,44]. Imbernón-Moya et al. used midazolam and general anesthesia (propofol) [34]. The PDT was not discontinued completely in any case due to severe pain, and only Biniszkiewicz et al. described the need for short breaks during the course of the procedure in some patients [26].

Erythematous lesions (*n* = 10), lasting up to a week after treatment, were another frequently observed side effect of PDT. Burning sensation (*n* = 9) and the presence of local swelling (*n* = 7) were reported slightly less frequently. Pruritus, paresthesia, urinary problems, erosions, and lubrication disorders were much less common.

Of note is the work of Osiecka et al., in which PDT was performed using green light to improve the tolerability of the procedure. In addition, a fractionated irradiation model was used, that is, two minutes of exposure was separated by a one-minute interval [35]. However, patients experienced pruritus of varying intensity as well as mild to moderate pain, which, in comparison with red light studies, for example, Zawiślak or Maździarz, implied poorer tolerance of the treatment [31,36].

### 3.5. Recurrences

VLS is a chronic disease, so relapses and exacerbations are part of its course. Romero et al. and Sotiriou et al., during follow-up after PDT (3–6 months and 3 months, respectively), observed mild symptoms of VLS, manageable with topical corticosteroids [27,29]. Osiecka et al. described the reappearance of erosions in one woman as early as 4 months after treatment and in two women after 6 months, accompanied by a burning sensation. In addition, during the 6 months after PDT, three patients reported weak pruritus, occurring mainly before menstruation [35]. Vano-Galvan’s case report presented a major recurrence 4 months after treatment, whereby the patient refused to continue PDT [30]. In Imbernón-Moya’s study, recurrence of lesions occurred between 3 and 9 months after treatment [34]. Li et al. observed a relapse in 2/10 patients after 6 months [41]. Zhang et al. reported recurrence in 3/30 women 6 months after treatment [44]. Furthermore, Olejek et al. noted pruritus in 8% of female patients during a 24-month follow-up [37].

However, recurrence of VLS was not described in other studies. It is worth mentioning that Maździarz et al. confirmed the absence of disease exacerbation in vulvoscopy performed one year after PDT [39].

## 4. Discussion

Currently, PDT has many applications in the treatment of a variety of skin disorders, mainly non-melanoma skin cancers [46]. Of note, this method in dermatology was used by Kennedy et al. for the first time [47]. The mechanism of PDT is based on the interaction of three elements: photosensitizer, the light of the appropriate wavelength, and oxygen. The purpose of this interaction is the production of cytotoxic reactive oxygen species that selectively destroy damaged tissue while leaving normal skin intact [47]. However, the exact mechanism of action of PDT in the treatment of VLS remains uncertain. It is thought to primarily target skin sclerosis as PDT has been shown to induce apoptosis of lymphocytes and keratinocytes and to alter the expression of both cytokines and metalloproteinases that play a role in skin remodeling [48]. However, Olejek et al. described the effect of PDT on the immune status of patients after the procedures, that is, a significant reduction in antinuclear antibody titers, which places PDT as a method with immuno-modulatory potential [37]. Moreover, Maździarz et al. presented the efficacy of PDT in concomitant infection with high-risk HPV types, which establishes PDT as a prophylactic method in cancer development [39]. According to Zielińska et al., treatment of VLS in the absence or confirmed presence of HPV infection is equally effective [45].

The first-line treatment for VLS is topical corticosteroids. When used chronically, these agents possess numerous side effects [49]. Moreover, the risk of VLS recurrence after their use is very high [50]. Noteworthy are the results of a single open-label study comparing ALA-PDT and the application of clobetasol propionate 0.05% ointment by 43 patients with VLS [51]. PDT treatments were performed four times every 2 weeks, and clobetasol was applied once daily for 8 weeks. It was found that clinical symptoms and subjective complaints improved in both groups. However, it was ALA-PDT that led to a higher clinical response rate and a longer remission period [51].

Biniszkiewicz et al. stated that PDT, because of its excellent cosmetic effect, lack of complications in the form of scarring, or changes in the structure and function of the treated tissues, should be performed as a therapy preceding more invasive procedures [26]. In view of the above and the lack of effect on reproductive capacity, PDT stands as a method also for the treatment of young women of childbearing potential [32]. PDT is considered an affordable method [26]. In addition, thanks to the possibility of using a form of a patch containing photosensitizer, it becomes an outpatient procedure, which is convenient and timesaving for patients [31].

Undoubtedly, VLS significantly affects the mental and physical health of female patients. Not only is the architecture and morphology of the vulva altered, but a wide range of subjective complaints are also experienced. The effectiveness of PDT in remission of symptoms of the disease, improvement of sexual life, and mood disorders allows making patients’ quality of life better. Based on the literature review, PDT for VLS apparently represents a promising therapeutic modality. In all of the papers, there was an improvement, reflected in the resolution of subjective and/or objective symptoms. Although some of the publications did not demonstrate the clinical efficacy of PDT in the treatment of VLS, none of them reported lesion exacerbation. However, a significant limitation in terms of assessing the efficacy of PDT was the paucity of controlled studies.

The histopathological findings considered demand a careful approach. The reported lack of improvement indicates the need for continued follow-up of patients for progression of lesions to malignancy. Of note, VLS, as an intractable dermatosis in some cases, will not change histologically. However, the reported evidence of apoptosis indicates the therapeutic effect of PDT [31].

In all reported cases, PDT was performed as the next step, not as first-line treatment. This emphasizes the role of PDT to be a therapeutic option for refractory lesions to the previous treatments. It may also be safely repeated a number of times as it is not associated with the development of resistance [41]. However, in order to reduce the number of procedures, to shorten the treatment period, and to avoid the side effects that accompany the interventions, it is worth considering the addition of a holmium laser treatment [42].

There was certainly a notable heterogeneity in the publications in regard to the photosensitizing substance and its retention length on the skin, the treatment parameters, their number, and the time intervals between them. In addition, the studies used different indices to assess clinical improvement. In our opinion, dermoscopy deserves special at-tention, which, as a non-invasive, available, and inexpensive method, allowed the evaluation of the therapeutic response at early stages [43]. In contrast, concordance was seen for the study cohort, which consisted of peri-menopausal women.

Interestingly, Declerq et al. recently reported a PDT protocol for VLS on the basis of a systematic review of the literature [52]. Patients should urinate before the procedure; then, the vulva is to be washed with 0.9% NaCl. The photosensitizer, 5% ALA, has to be applied under the occlusion with a margin of 1 cm. The incubation period is supposed to be 3 h, and red light of 590–760 nm, at a dose of 120 J/cm^2^, and an intensity of 204 mW/cm^2^ should be used for irradiation. In addition, the authors suggested the practice of blue light photodiagnostics and xylocaine or water spray alleviate side effects during the procedure [52].

## 5. Conclusions

The results of this systematic review of the literature indicated that PDT is a valuable therapeutic modality in the treatment of VLS, especially those which are refractory to current treatment. It is undoubtedly a high-efficacy method, particularly in terms of resolution of subjective symptoms, which is reflected in an improvement in the quality of life of treated women. This method has a high safety profile and in most cases is associated with the development of mild side effects. Treatments may be repeated several times at no risk of resistance development. Moreover, PDT is characterized by a lack of negative effects on women’s reproductive potential and high patient satisfaction with the treatments. The good cosmetic effect of PDT and its potential for cancer prevention are also worth mentioning. However, the lack of histopathological remission should prompt a long-term observation of patients for cancer progression.

## Figures and Tables

**Figure 1 jcm-10-05491-f001:**
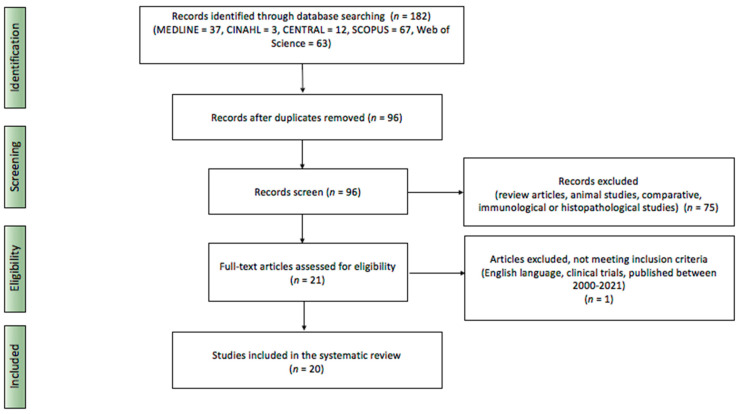
Literature search based on PRISMA protocol.

**Table 1 jcm-10-05491-t001:** Studies included in this systematic review.

No.	Study	Type of Study	Number of Patients/Patients’ Age	Previous Treatment of VLS	Photosensitizer: Type/Time of Incubation	PDT Characteristic: Light/Wavelength/Power Density	Number of Treatment/Time Interval	Outcome	Adverse Effect
1	Biniszkiewicz et al. (2005) [26]	Prospective cohort	24 women/ 58 years on average	Not reported	20% 5-ALA/115 min	Coherent/630 nm, 180 J/cm^2^, 700 mW/cm^2^	3–6 therapy cycles/14-day interval	-remission of subjective symptoms in 23 patients -maintenance of the subjective symptoms in one patient	minimal local toxicity: vulvar erythema
2	Romero et al. (2007) [27]	Case report	One woman/ 61 years	Surgical treatment, topical corticosteroids, topical tacrolimus 0.1%, hydroxychloroquine 200 mg/day, oral prednisone 0.5 mg/kg/day	20% 5-ALA/2 h	Noncoherent 633 nm red light, 30 J/cm^2^, 80 mW/cm^2^	Two treatments/1-month interval	-almost total reduction of subjective symptoms; -marked improvement of objective symptoms; -improvement of sexual life	Moderate pain during irradiation
3	Sotiriou et al. (2008) [28]	Case series	10 women/ 54.6 years on average	Topical corticosteroids, pimecrolimus ointment	20% 5-ALA/4 h	Noncoherent 570–670 nm red light, 40 J/cm^2^, 80 mW/cm	Two treatments/2-week interval	-minor improvement of clinical signs in nine patients; -no improvement in one patient; -remission or reduction of subjective symptoms in all of the patients	Burning and stinging sensation during treatment; erythema up to 1 week after irradiation
4	Sotiriou et al. (2008) [29]	Case series	Five women/ 61.4 years on average	Topical corticosteroids, tacrolimus ointment 0.1%	20% 5-ALA/3 h	Noncoherent 570–670 nm red light, 40 J/cm^2^, 80 mW/cm^2^	One session	-significant reduction of subjective signs in all patients; -no resolution of histopathological features; -mean duration of reduction of symptoms 4.6 months	Burning sensation during the procedure, local erythema 3–5 days after therapy
5	Vano-Galvan et al. (2008) [30]	Case report	One woman/ 68 years	Topical 0.1% halcinonide once daily, tacrolimus 60.1% ointment, oral prednisone 0.5 mg/kg/day	MAL/2 h	Coherent, 585 nm, 7 mm, 6 ms, 9 J/cm^2^	Three cycles/a one-month interval	-marked clinical improvement, -almost total reduction of subjective symptoms	Intense pain during the procedure
6	Zawiślak et al. (2009) [31]	Prospective cohort	Eight women/ Age not reported	Not reported	Bioadhesive patch system with ALA/4–6 h	Noncoherent 630 nm red light, 100 J/cm^2^	Two sessions/2–15-week intervals	-marked decrease of subjective symptoms; -no resolution of histopathological features	Intense pain during first 3 min of irradiation; post-treatment pain lasting for at least 24 h
7	Skrzypulec et al. (2009) [32]	Prospective cohort	37 women/ 50–70 years (59.98 years on average)	Not reported	5% ALA/4–5 h	Coherent 635 nm, 80 J/cm^2^, 40–70 mW/cm^2^	Six cycles/2-week interval	-reduction in the severity of symptoms in 28 patients; -no negative influence on sexual life; -beneficial effect on depressive disorders	Lubrication disorders
8	Osiecka et al. (2012) [33]	Case report	One woman/ 30 years	Clobetasol propionate, tacrolimus	20% 5-ALA/4 h	Noncoherent 630 nm red light, 150 J/cm^2^, 100 mW/cm^2^	a total of six sessions; the first two separated by 4 weeks; the 3rd performed 6 months after starting treatment; the 4th after another 6 weeks; no information about the 5th and 6th sessions is available	-complete remission of objective and subjective symptoms, except few days in the perimenstrual period	Marked itching during the first treatment, burning within 24 h after the first irradiation
9	Imbernón-Moya et al. (2016) [34]	Case series	Eight women/ seven women > 60 years One woman: 38 years	Topical corticosteroids, topical calcineurin inhibitors	MAL/3 h	Noncoherent 630 nm red light, 37 J/cm^2^, 70 mW/cm^2^	1–3 treatments/6–12 months interval	-significant improvement in subjective symptoms and quality of life in all cases; -lack of improvement in clinical presentation in all cases	Mild erythema, edema, burning, urinary problems
10	Osiecka et al. (2017) [35]	Prospective cohort	11 women/ 30–66 years (48 on average)	Topical corticosteroids, estrogens, topical calcineurin inhibitors	20% 5-ALA/5 h	Noncoherent 540 ± 15 nm green light, 62.5 J/cm^2^, 85 mW/cm^2^; fractionated—2 min irradiation, then 1 min pause	Three treatments/2-week interval	-complete resolution of objective symptoms in 5/5 patients 2 months after PDT; -complete resolution of subjective symptoms 2 months after PDT in 9/11 women, one remaining in moderate intensity, the other one in low intensity	Itching as the main symptom, weak or moderate pain, mild edema, and erythema
11	Maździarz et al. (2017) [36]	Prospective cohort	102 women/ 19–85 years (55.08 on average)	Topical corticosteroids	5% 5-ALA with 2% DMSO/3 h	Noncoherent 590–760 nm, 120 J/cm^2^, 204 mW/cm^2^	10 applications/one-week interval	-complete or partial remission in 87% of patients, -decrease in the number of objective signs (improvement rate 100%—70% in 60.78% of patients, around 50% in 16.67% of patients, 30% in 9.8%, less than 30% in 12.75%)	Paresthesia during irradiation in 39 patients, in 12 patients swelling for a few hours
12	Olejek et al. (2017) [37]	Prospective, controlled cohort	100 women/ 57 years on average in the first group (*n* = 40) and 58.5 years on average in the second group (*n* = 60)	Not reported	10% ALA with 20% DMSO/3 h	First group: coherent 630 nm red light Second group: 580–1400 nm 100 J/cm^2^; 40–80 mW/cm^2^	10 applications/one-week interval	-significant reduction of subjective symptoms in 92% patients, -in 8% of patients’ symptoms of the same or worse intensity	No visible side effects
13	Lan et al. (2018) [38]	Case series	10 women/ 51 years on average	Topical corticosteroids, topical calcineurin inhibitors, cryosurgery	10% 5-ALA/3 h	Noncoherent 635 ± 15 nm red light, 100 J/cm^2^, 100 mW/cm^2^	Three sessions/2-week interval	-resolution of subjective symptoms in 10 patients, -complete resolution of sexual dysfunction; -improvement of quality of life; -significant decrease of lesion size; -no recurrences during the observation period	Short-term pain, burning, erythema, and edema during and after irradiation
14	Maździarz (2019) [39]	Prospective cohort	Two women/ 22 and 23 years	Topical corticosteroids	5% 5-ALA with 2% DMSO/3 h	Noncoherent 590–760 nm, 120 J/cm^2^, 204 mW/cm^2^	10 applications/one-week interval	remission of vulvar lesions and negative HPV DNA results in one patient	Short-term pain and burning sensation
15	Zhang (2020) [40]	Case series	30 women/ 48.23 years on average	Topical corticosteroids, vitamin E, Haijisin	20% 5-ALA/3 h	Noncoherent 631–635 nm red light, 60–90 mW/cm^2^	Three sessions/ 2-week interval	-total resolution of pruritus in 25 patients, improvement in three patients, -complete resolution of pain in 28 patients, mild to moderate pain in two patients -total resolution of sexual dysfunction in 26 patients, moderate to severe sexual intercourse persisting in 4 patients	Short-term pain, burning, erythema, and edema
16	Li et al. (2020) [41]	Prospective cohort	10 women/ 35.4 years on average	Topical corticosteroids	20% 5-ALA/3 h	Coherent 635 nm red light, 80 J/cm^2^, 80 mW/cm^2^	4–9 sessions, depending on the condition	-significant reduction in objective and subjective symptoms of VLS and improvement in quality of life -no recurrence of lesions 3 months after PDT -recurrence of lesions 6 months after PDT in two patients	-mild to moderate pain in eight women undergoing PDT -burning sensation, swelling, erythema in six women, lasting up to 5 days after irradiation
17	Cao et al. (2020) [42]	Case report	One woman/ 72 years	Clobetasol propionate 0.5% cream	10% 5-ALA/3 h	Noncoherent 635 nm red light, 100 J/cm^2^, 200 mW/cm^2^	Three treatments at 2-week intervals, then after one month, a holmium laser treatment in combination with the last PDT	-VLS areas reduced and thinned after three PDT treatments -almost complete remission after last combined laser + PDT treatment -relief of subjective symptoms -no recurrence of lesions after 1 year of follow-up -satisfaction of patients with the treatment	Mild swelling and erythema after each treatment, moderate pain
18	Liu et al. (2021) [43]	Prospective cohort	24 women/ 21–61 years (45 years on average)	Not reported	20% 5-ALA/3 h	Noncoherent 633 nm red light, 60 mW/cm^2^	Six treatments/2-week intervals	-significant remission of clinical signs -gradual alleviation of subjective symptoms with subsequent PDT treatments -marked improvement of the dermoscopic features	-in 19 cases, transient complaints of pain (<24 h) -in seven cases, erosions healed within one week after PDT
19	Zhang et al. (2021) [44]	Prospective cohort	30 women/ 48.2 years on average	Topical corticosteroids, vitamin E cream	20% 5-ALA/3 h	Noncoherent 635 nm red light, 100–150 J/cm^2^, 60–90 mW/cm^2^	Three sessions/2-week interval	-significant reduction of objective signs -improvement of histopathological findings -significant reduction of sexual dysfunctions, considerable improvement of patients’ quality of life -recurrence of lesions in three patients at 6 months follow-up	-pain and burning sensation gradually subsiding 3 to 48 h after the procedure -slight erythema and swelling for up to 4 days after PDT
20	Zielińska et al. (2021) [45]	Prospective cohort	73 women/ 9–81 years (54.1 years on average)	Not reported	5% 5-ALA with 2% DMSO/2 h	Noncoherent 630 nm, 120 J/cm^2^ 204 mW/cm^2^	a full cycle of 10 treatments once a week; if necessary, the cycle was repeated after 3 months (one cycle: 37 women, two cycles: 30 women, three cycles: six women)	-resolution of subjective symptoms in all patients -histopathological remission in two patients -clinical remission without histopathological remission in 55 patients -no significant influence of HPV infection on the number of procedures -no relevant correlation between duration of clinical remission and HPV status	32 patients reported paresthesia during PDT that resolved after treatment

5-ALA—5-Aminolevulinic acid, MAL—methyl aminolevulinate, DMSO—dimethyl sulfoxide. Included subjective symptoms such as pruritus, burning, pain. Included objective symptoms such as erythema, erosions, ecchymoses, telangiectasias, fissures, lichenification with hyperkeratosis, atrophic lesions, size of lesions, pigmentation of lesions, purpuric lesions, and excoriation.

## Data Availability

Data sharing not applicable.

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
