# Peer review of "Photodynamic Therapy in the Treatment of Vulvar Lichen Sclerosus: A Systematic Review of the Literature"

_jcm, 2021, doi:10.3390/jcm10235491_

Round 1

Reviewer 1 Report

  1. The introduction is a bit too long, to my eyes. The most important aspect here is the treatment of the disease. What are the options? What is the role of PDT according to the current guidelines? Besides that, maybe a short description of the disease, its prevalence and the importance of treatment due to high impact on quality of life and secondary malignancy would be enough. 
  2. The Methods - not the Results - should include the criteria of inclusion (English language publications, clinical trials, publications 109 from 2000-2021) and exclusion (review articles, animal studies, as well as comparative, immunological, or histopathological studies).
  3. In the text of the Results section I don't have any comments. Congratulations for the organization of the text.
  4. In the "Type of study" column in Table 3 it would be useful to add which studies are prospective cohorts and which retrospective.
  5. In the Discussion, it is very important to state that there are very few controlled studies which is a great limitation to consider effectiveness. Many studies find that objetive mesures do not improve with PDT and the main outcomes that are found to improve are subjective mesures like pain, pruritus and quality of life. These symptoms are especially prone to be modified by placebo. Taking into account the questionable benefit, it is very important to consider safety. Pain during the procedure is, in my opinion, an important limitation. 

Author Response

Response to Comments from REVIEWER #1

Dear Reviewer,

We would like to thank you for careful and thorough reading of this manuscript and for the thoughtful comments and constructive suggestions, which help to improve the quality of this manuscript. As below, on behalf of my co-authors, I would like to clarify some of the points raised in this review.

Comment 1:

The introduction is a bit too long, to my eyes. The most important aspect here is the treatment of the disease. What are the options? What is the role of PDT according to the current guidelines? Besides that, maybe a short description of the disease, its prevalence and the importance of treatment due to high impact on quality of life and secondary malignancy would be enough.

Response:

As suggested, we have shortened the introduction slightly. We have left information on the prevalence of the disease, its clinical presentation and its impact on patients' quality of life. In addition, we have drawn more attention to the treatment of VLS, taking into account evidence-based recommendations. 

However, due to comments of other Reviewer, we were obligated to add additional information in this section (eg. that in the course of lichen sclerosus involvement of other body regions as well as the genital area in men may occur, and about the phototoxicity of the photosensitizer applied during PDT) thus, ultimately, it did not considerably shorten in revised version of the manuscript. We hope it will be acceptable for the Reviewer.

Comment 2:

The Methods - not the Results - should include the criteria of inclusion (English language publications, clinical trials, publications 109 from 2000-2021) and exclusion (review articles, animal studies, as well as comparative, immunological, or histopathological studies).

Response:

We thank the Reviewer for drawing attention to the need to include this important information. In revised manuscript we have changed this part.

Comment 3:

In the text of the Results section, I don't have any comments. Congratulations for the organization of the text

Response:

We are grateful and delighted to receive this remark. Thank You.

Comment 4:

In the "Type of study" column in Table 3 it would be useful to add which studies are prospective cohorts and which retrospective.

Response:

We agree with the suggestion from the Reviewer. We have added this information in the Table. However, none of the papers mentioned were retrospective.  

Comment 5:

In the Discussion, it is very important to state that there are very few controlled studies which is a great limitation to consider effectiveness. Many studies find that objetive mesures do not improve with PDT and the main outcomes that are found to improve are subjective mesures like pain, pruritus and quality of life. These symptoms are especially prone to be modified by placebo. Taking into account the questionable benefit, it is very important to consider safety. Pain during the procedure is, in my opinion, an important limitation.

Response:

We thank the Reviewer for careful and thorough reading of this manuscript. We undoubtedly agree with the Reviewer about the opinion that too few controlled studies do not allow to assess the effectiveness of the discussed therapeutic method. Moreover, this method does not allow for complete clinical remission in terms of objective symptoms. We have included a relevant part in the manuscript.

In our opinion, the relief of subjective complaints in the course of VLS is crucial and represents an undoubted advantage of photodynamic therapy. Furthermore, based on a review of the literature, PDT is generally a well-tolerated method and in none of the papers mentioned did pain led to withdrawal of treatment. However, each patient should receive information about potential pain before treatment (according to evidence-based consensus)

Reviewer 2 Report

An interesting systematic review about the use of photodynamic therapy in vulvar lichen sclerosis may be also useful to various clinicians. I found it very concise, and eligible to be published after minor revisions.

page 1 line 28 you should add: "lichen sclerosus may also interest other body areas, and the genital area of the opposite sex" and cite an article such as: https://doi.org/10.34172/jlms.2021.61

Page 2 line 90 you should add "photodynamic therapy is a treatment consisting in the use of a photosensitizing chemical substance to cause phototoxicity" and cite an article such as: doi: 10.23736/S0392-0488.19.06392-2.

Thank You

Author Response

Response to Comments from REVIEWER #2

Dear Reviewer,

We would like to thank you for careful and thorough reading of this manuscript and for the thoughtful comments and constructive suggestions, which help to improve the quality of this manuscript. As below, on behalf of my co-authors, I would like to clarify some of the points raised in this review.

Comment 1:

An interesting systematic review about the use of photodynamic therapy in vulvar lichen sclerosis may be also useful to various clinicians. I found it very concise, and eligible to be published after minor revisions.

Page 1 line 28 you should add: "lichen sclerosus may also interest other body areas, and the genital area of the opposite sex" and cite an article such as: https://doi.org/10.34172/jlms.2021.61

Response:

We are grateful for this suggestion. In fact, suggested reference is valuable and we have added it in our revised manuscript. It has position [4] in the reference list. 

Comment 2:

Page 2 line 90 you should add "photodynamic therapy is a treatment consisting in the use of a photosensitizing chemical substance to cause phototoxicity" and cite an article such as: doi: 10.23736/S0392-0488.19.06392-2.

Response:

We thank the Reviewer for the suggestion. We have included the proposed reference in our revised manuscript. It has position [26] in the reference list.